# Breast-Specific Gamma Imaging: An Added Value in the Diagnosis of Breast Cancer, a Systematic Review

**DOI:** 10.3390/cancers14194619

**Published:** 2022-09-23

**Authors:** Maria Silvia De Feo, Marko Magdi Abdou Sidrak, Miriam Conte, Viviana Frantellizzi, Andrea Marongiu, Flaminia De Cristofaro, Susanna Nuvoli, Angela Spanu, Giuseppe De Vincentis

**Affiliations:** 1Department of Radiological Sciences, Oncology and Anatomo-Pathology, Sapienza University of Rome, 00161 Rome, Italy; 2Unit of Nuclear Medicine, Department of Medical, Surgical and Experimental Sciences, University of Sassari, 07100 Sassari, Italy

**Keywords:** breast specific gamma imaging, ultrasound, mammography, MRI, US, MRSI, BSGI and ultrasound, BSGI and MRI, BSGI and MMG

## Abstract

**Simple Summary:**

Breast-specific gamma imaging represents an emergent instrument for breast cancer detection. We selected on Medline articles published from 1995 to 2022 that compare various imaging modalities with breast-specific gamma imaging. The aim of this paper was to assess if this imaging method is a more valuable choice in detecting breast malignant lesions compared to morphological counterparts such mammography, ultrasound, and magnetic resonance imaging in terms of specificity, sensibility and positive and negative predictive value. At the cost of a major radiology burden, breast-specific gamma imaging is more specific, with a sensibility comparable to magnetic resonance imaging and higher than ultrasonography and mammography.

**Abstract:**

Purpose: Breast cancer is the most common solid tumor and the second highest cause of death in the United States. Detection and diagnosis of breast tumors includes various imaging modalities, such as mammography (MMG), ultrasound (US), and contrast-enhancement MRI. Breast-specific gamma imaging (BSGI) is an emerging tool, whereas morphological imaging has the disadvantage of a higher absorbed dose. Our aim was to assess if this imaging method is a more valuable choice in detecting breast malignant lesions compared to morphological counterparts. Methods: research on Medline from 1995 to June 2022 was conducted. Studies that compared at least one anatomical imaging modality with BSGI were screened and assessed through QUADAS2 for risk of bias and applicability concerns assessment. Sensitivity, specificity, positive and negative predictive value (PPV and NPV) were reported. Results: A total of 15 studies compared BSGI with MMG, US, and MRI. BSGI sensitivity was similar to MRI, but specificity was higher. Specificity was always higher than MMG and US. BSGI had higher PPV and NPV. When used for the evaluation of a suspected breast lesion, the overall sensitivity was better than the examined overall sensitivity when BSGI was excluded. Risk of bias and applicability concerns domain showed mainly low risk of bias. Conclusion: BSGI is a valuable imaging modality with similar sensitivity to MRI but higher specificity, although at the cost of higher radiation burden.

## 1. Introduction

Breast cancer is the leading cause of malignancy in women, followed by lung cancer, and the second most common cause of cancer death in the US. In recent years, a slow and steady rise in the detection of breast cancer has been seen, although mortality has been reduced thanks to an earlier diagnosis and treatment, but also sensibilization campaigns and better patient management [1,2]. Mammography (MMG) is the most used screening method used to explore anatomic changes. Its sensitivity is influenced by breast density and ranges from 75 to 85% in women with low-fat tissue to 42 to 68% in women with high-fat tissue [3,4,5]. Other limitations are represented by previous surgery or radiation therapy as well as architectural distortion, which might lead to unnecessary biopsies [6,7]. In order to overcome these pitfalls, other imaging techniques are used, such as ultrasound (US) and magnetic resonance imaging (MRI). Hand-held ultrasound (HHUS) has its flaws such as operator-dependency, small field of view (FOV), and the lack of standardization. Nevertheless, it has a small increase in sensitivity when coupled to mammography at the cost of a reduction in specificity. Automated Breast Ultrasound (ABUS) was developed to reduce operator dependency, although no specificity was observed [8]. MRI, although radiation-free, is still burdened with renal toxicity due to contrast enhancement [8]. More recently, Magnetic Resonance Spectroscopy Imaging (MRSI) has been shown to be very sensitive, although it does not have brilliant specificity [9]. The improvement of imaging devices results in a better sensitivity, specificity, and positive predictive value (PPV) [10,11]. These results have been recently reached in nuclear medicine imaging thanks to Breast-specific Gamma Imaging (BSGI). This is a radio-imaging technique used to precisely detect breast cancer lesions, by applying dedicated detectors, in tissues of variable density [12]. The main radiopharmaceutical (RP) used for BSGI is Technetium 99metastable (99mTc) methoxyisobutyl isonitrile (Sestamibi-MIBI) [13]. It is a cationic and lipophilic molecule with a great affinity for high metabolism and therefore for mitochondria-rich cells. Its distribution is dependent on blood flow, and it is influenced by mitochondrial membrane potential. These characteristics make MIBI a good tracer for tumor lesions imaging and tissues with high blood flow and high metabolism. A lack of uptake may be seen in high concentrations on efflux proteins such as P-glycoprotein and multidrug resistance-related protein 1 [14]. The recommended activity of 99mTc-sestamibi is 370–740 MBq. More recent studies showed how a 2.5-fold dose reduction in activity was possible without a significant impact on sensitivity and specificity, using both NaI and CZT detectors. For 296 MBq i of administered activity, the effective whole-body dose is 2.5 mSv and 0.019 mGy/MBq for the breast, while for two-projection mammography the effective dose delivered to the breast is 0.5 mSv. It is worth noting how the estimations of radiation-related cancer with a dose below 100 mSv are almost zero [15,16]. With an injected activity of 740–1100 MBq, critical organs are represented by the large intestine and by secretory organs such kidneys, bladder and gallbladder wall. The breast has the lowest dose equal to 2 mGy. The effective dose is estimated to be 5.9–9.4 mSv. Considering the properties of Sestamibi that adheres to the plastic syringe, a 20% to 30% of activity is maintained in the syringe after injection, thus an administered activity is inferior. For example, the effective administered activity of a 296 MBq contained in the syringe is about 240 MBq [17]. A BSGI conducted with activities below 300 MBq demonstrated a superior cancer detection rate than mammography alone, but it does not match the benefit-to-radiation risk ratio of screening mammography [18]. 99mTc-tetrofosmin is another radiotracer that is used for BSGI and, like MIBI, finds its main application in myocardial imaging. Its uptake mechanism is very similar to that of MIBI. It is a metabolism-dependent process that does not imply cation channel transport but most likely diffusion of the cation through mitochondrial membranes. Sensitivity and specificity for BSGI with tetrofosmin overlap the sensibility and specificity of BSGI conducted with MIBI. The only advantage of tetrofosmin could be found in its preparation since it does not require boiling before usage (as MIBI) [19,20]. Newer RPs are still under development and are pending for FDA approval. Maraciclatide (99mTc-NC100692) is a synthetic cyclic peptide with strong affinity for tissue characterized by high angiogenesis and pregulated integrins such as αvβ3. Thus, its target is the integrin’s receptors expressed on endothelial cells. In 2017, O’Connor et al. published data regarding BSGI using MIBI and BSGI with Maraciclatide on 39 women with known or suspected breast lesions, although no significant difference was found between the two RPs in detecting malignant breast lesions (AUC 0.83 and 0.87 for MIBI and Maraciclatide, respectively). They showed comparable uptake in breast tumors [21]. In another paper, recently published specific peptides with high cell binding properties were reported. Ahmadpour et al. studied MCF-7 cell line, which is an epithelial luminal cell line expressing progesterone and estrogen receptors. This cell line is isolated from bone-metastatic women with pleural effusion. They demonstrated that 99mTc-HYNIC-(tricine/EDDA)-FROP showed relevant uptake in the MCF-7 cell lines, compared to other normal and cancerous cell lines, figuring as a promising probe for these malignancies [22,23]. The aim of this systematic review was to assess if BSGI is a more valuable choice in detecting breast malignant lesions compared to morphological counterparts.

## 2. Materials and Methods

The research was conducted on Pubmed starting from January 1995 to June 2022 and drawn up according to PRISMA guidelines. The protocol has not been registered. The following keywords were applied: “BSGI” or “breast specific gamma imaging” and “ultrasound” or “MRI” or “magnetic resonance imaging” or “MMG” or “mammography”. Studies that made a comparison between at least two imaging modalities were tabled and sensibility, specificity, PPV and negative predictive value (NPV) were reported. Papers on BSGI only or with comparison between groups of imaging modalities were also considered for this paper. English language was mandatory. Quality Assessment of Diagnostic accuracy Studies-2 (QUADAS-2) was the mean for assessing the diagnostic validity of eligible studies.

## 3. Results

Through these keywords, 1196 articles were screened. After excluding duplicates, case reports, reviews, trials, non-English articles, and non-related papers, a total of 15 comparison studies between BSGI were collected (see Figure 1). Among all imaging modalities, MRI displayed the highest sensitivity, ranging from 83.3% to 100%. Specificity was low from 25% to 69.7%. Mammography’s sensitivity ranged from 53.6% to 93.64%, while specificity was from 28% to 90.66%. Ultrasound had high sensitivity, from 82.1% to 99%, but lower specificity, from 19.8% to 87.09%. BSGI had high sensitivity, along with MRI, from 68.6% up to 95.45%. BSGI has a specificity between 56% and 90.93%. PPV and NPV, when reported, were almost always higher for BSGI than for the compared exam. The inclusion of BSGI as a breast lesion diagnostic tool increased overall sensitivity. The differences in sensitivity, specificity, PPV, NPV between BSGI and MRI in the included studies is summarized in Table 1, while the differences between BSGI and mammography in Table 2. In Table 3, BSGI and ultrasound are compared. Methodological quality assessment was satisfactory. Out of 105 domains, only 4 were high risk, and 2 other domains were unclear. Most of the evaluated domains were at a low risk of bias. High risk domains were a concern in one study, as the authors selected a group of patients whose anatomical abnormalities were most likely not due to cancer. Overall, low risk assessment is probably due to a conventionally standardized procedure in breast cancer management, going through imaging, biopsy and surgery and with no concerns for applicability whatsoever (See Table 4).

## 4. Discussion

Liu et al. evaluated 390 women with suspected breast lesions with BSGI, 235 of whom underwent MRI as well. MRI had the highest sensitivity but lowest specificity among all examinations and higher costs and longer imaging time compared to mammography and ultrasound. BSGI had higher positive and negative predictive value (PPV and NPV), as well as the highest specificity and the second-highest sensitivity. BSGI was found to be highly sensitive in detecting and diagnosing ductal carcinomas in situ (DCIS) [12]. In 2021 the same group studied Sestamibi BSGI in women undergoing neoadjuvant chemotherapy (NAC) to evaluate tumor staging and NAC response, since MRI yields high false-positive rates and it may well overestimate tumor size, leading to unwanted surgery. A slightly lower sensitivity for BSGI was confirmed with respect to a greater specificity compared to MRI [24]. In 2012, Lee et al. compared BSGI to mammography and ultrasound, subdividing the study group of 471 women into under and over 50 years of age (249 and 222, respectively). Sensitivity and specificity were overall higher in the over 50 group for all the imaging modalities, probably due to the higher breast density in the under 50 group, leading to more false negative results in the latter group [33]. In 2020, Liu et al. compared BSGI with MMG and US in women with Breast Imaging Reporting and Data System 4 (BI-RADS 4) lesions in 177 women retrospectively. Lesion sizes ranged from 3 to 74 mm. BSGI sensitivity for lesion <1 cm was comparable to US, with higher specificity compared to MMG and US (88% vs. 40% and 64%, respectively). In lesions with dimensions greater than 1 cm, specificity dropped to 71.4%, still higher than US 45.7%. Among luminal lesions, HER2+ and triple negative, sensitivity overlapped between the three modalities [34]. Keto et al. compared BSGI and MRI in patients with ductal carcinoma-in situ (DCIS), as it is usually detected through breast calcification seen on MMG, although reported sensitivity in the literature fluctuates. A total of 18 women with recently discovered DCIS who underwent MRI and BSGI were grouped, and MRI identified 17 lesions while BSGI identified 16 women positive for malignant lesions (94% vs. 89%). No significant difference was found. The non-identified lesion on both imaging modalities had dimensions of 0.2 cm [25]. DCIS turned out to be the most detected malignancy through BSGI. In 2007, Brem et al. reported data on the sensitivity for BSGI in comparison to MMG and MRI for DCIS. It was equal to 92%, 82% and 88%, respectively. Another paper on multiple malignancies reported lower specificity compared to other imaging modalities (59.5%). The highest sensitivity was reported for lesions greater than 11 mm (100%) [26,32]. In 2016, Brem et al. showed an increase in diagnostic accuracy when BSGI was added to the annual screening MMG in inconclusive exams conducted in women with dense breasts. This proved to be a limitation of MMG, demonstrating how improved overall sensitivity by 1.7% was reached when BSGI was included in the screening program [37]. Other authors’ results followed the overall pattern comparing BSGI to MMG and US, with higher overall sensitivity and specificity found for BSGI in BI-RADS 4. These results were repeatable when women with dense breasts were considered, especially specificity-wise (81.3% vs. 19% and 50% for BSGI, US and MMG, respectively). BSGI showed excellent results in dense breasts and lesions with dimensions less than 1 cm with only suspicious microcalcifications [35]. Semi-quantitative BSGI analysis was also applied to compare BSGI to MMG, MRI and US in BI-RADS 4 and 5 lesions, by dividing the number of counts in the BSGI lesion by the ipsilateral background parenchyma. Mean value (counts in BSGI lesion) was 4.27 and 2.37 for benign ones, with a cut-off of 3.04. Relative lesion size was not significantly different (18.42 cm vs. 17.88 cm for malignant and benign lesions, respectively) [27]. Yu et al. in 2016 examined 166 women who had undergone four imaging modalities. MRI had the highest sensitivity and BSGI the highest specificity, in line with other researchers. BSGI had the highest sensitivity, followed by MRI. Tumor to normal tissue ratio cut-off value was found at 1.82 [28]. Previously, Kim et al. also demonstrated the same sensitivity and specificity-wise for MRI and BSGI in women with dense breasts (>50%) [30]. Later, they evaluated MRI and BSGI performance in women after NAC. NAC response was determined by final surgical pathology, with a slight edge for BSGI (kappa values 0.47 vs. 0.41) [31]. Another study evaluated calcified and non-calcified DCIS with MRI and BSGI. Far greater sensitivity for MRI than BSGI in both sub-groups with an overall MRI sensitivity of 91.4% vs. 68.6 for BSGI was highlighted [29]. In 2012, Park et al. studied the clinical utility of BSGI in single and dual-phase imaging, acquired immediately after injection of MIBI and one hour later. Imaging was carried out on patients with BI-RADS from 1 to 5 categorized in US. Sensitivity decreased after dual imaging (77% to 69%), although not significantly, while specificity and PPV increased. As MIBI uptake in benign conditions decreases over time, resulting in asymmetrical mild and nodular uptake on early imaging, late imaging is supposed to confirm or exclude malignancy [38]. The same group in 2014 published a paper dividing breast cancer into BSGI+ and BSGI− lesions. Overall sensitivity was 85.7%, 90.7% for lesions greater than 1 cm, and 55.6% for sub-centimetric lesions, with the smallest lesion of 0.4 cm identified on BSGI only. No correlation was found between BSGI positivity and extensive intraductal component, Estrogen Receptor (ER), Progesterone Receptor (PR), ERBB2, Ki67, p53 and nuclear grade [39]. Again, in 2018, the aforementioned trend was confirmed on 89 invasive breast cancers imaged by BSGI in a study conducted by the same group. Reported sensitivity aligned with the one stated in the 2014 study, as well as for lesions over and below 1 cm. However, the sensitivity was lower than in other studies, probably due to the inclusion of only invasive breast cancers [40]. BSGI in Mucinous Breast Carcinoma (MCB) and Pure Mucinous Breast Carcinoma (PMBC) has been less studied compared to DCIS. It is harder to detect as the mucinous component takes over and less blood volume/mitochondrial density is to be found since a lesser uptake was seen. It might be responsible for misinterpretation since it could be mistaken for a benign lesion [41]. In the 2020, another Chinese group evaluated sensitivity and specificity for women tested with BSGI and MMG group and for MMG and US group. The population was characterized by 364 women with suspicious lesions and was heterogeneous. Only young women who are more likely to have denser breasts were selected. The MMG and US group had sensitivity equal to 90.4%. The BSGI and MMG group’s sensitivity was 93.6%. The highest difference between the two groups was the different representation of HER-2+ type (86.21 for MMG + US group, 96.55 for BSGI + MMG group) and the number of tumor lesions was smaller than 1 cm (73.33 for MMG + US group, 80 for BSGI + MMG group). AUC was 0.9 in the MMG + BSGI group and 0.93 in the MMG + US group [42]. In another study, Chung et al. evaluated the combination of BSGI and MMG compared with the group who had undergone MMG and US. The latter combination had higher sensitivity, although not significant, while the first had higher specificity, which was significant. PPV, NPV, and AUC were higher when BSGI was included. AUC did not change when patients were injected with 370 or 740 MBq of 99mTc-MIBI [43]. Sensitivity according to breast density is still controversial, as some data showed that sensitivity decreases when breast density increases, while others found no significant difference [44]. No consensus was reached in stating if BSGI is able to identify or predict ER, PR, HER-2, and Ki-67 expression. The value of lesion/normal tissue uptake has been demonstrated not to be related to receptor expression or histologic grade. It seemed influenced by infiltration degree and tumor size [36]. Another limitation for BSGI could be an increased background activity in women with dense breasts displayed in MMG and increased background enhancement on MRI, as it may interfere with lesion detection. BSGI could also be affected by the menstrual cycle, as women should be imaged between the 7th and the 14th day of the cycle [45]. However, the use of MRI does not seem to decline even though it underperformed compared to BSGI. Estimated in the United States, if every newly breast cancer diagnosed woman had to undergo MRI instead of BSGI, the costs would be more than 500M USD greater [46]. Nevertheless, the benefit-to-radiation risk ratio in annual screening for the combination of BSGI and MMG is similar than BSGI alone. BSGI risk from ionizing radiation is eight times that of MMG in women between 40 and 49, and about thirty times that of MMG in women between 70 and 79 [18].

Moreover, mean costs for mammography has been estimated around USD349 ($493), US around USD132 ($134) and biopsies USD1938 ($2343) in the USA between 1 January 2012 and 30 June 2014. The annual diagnostic breast cost of each procedure was USD3.05 billion, USD0.92 billion, and USD3.07 billion, respectively. Breast biopsy is burdened by a false positive rate in the USA, estimated at 71.0% with an annual false positive cost of USD 2.18 billion. In addition, 49.4% of patients underwent a second diagnostic procedure, 20.1% a third diagnostic procedure, and 10.0% fourth diagnostic procedure, with an increase in annual costs [47]. An analysis of the utilization patterns and associated costs of breast imaging and diagnostic procedures after screening mammography found that the total cost for BSGI amounts to USD 63,750, while the charge for false positive for BSGI is USD 8500 without biopsy. The cost of MRI is USD 253,575, while the cost of false positive excluding biopsy procedures amounts to USD 30,429 [46]. As can be seen, the gap between the two procedures is $189,825 in favor of BSGI, while the gap between the false positive cost is $29,929 in favor of BSGI. A more specific and sensible procedure could lower health care expenditures. In fact, it is evident that the higher sensibility and specificity of BSGI lowers the total costs for the national care system.

As BSGI finds its place in traditional nuclear medicine, PEM (Positron emission mammography) or MAMMI-PET (MAMmography and Molecular Imaging—Positron Emission Tomography) is its counterpart in PET imaging. While PET focuses on whole body imaging, PEM centers on breast imaging, coming at a lower cost and a higher spatial resolution. PEM uses two positron detectors mounted on a mammography gantry that are placed on both sides of the breast. [48]. FDG-PET is reported to have low sensitivity in screening breast cancer, lower than mammography and physical examinations together, [49]. A study that compared PET, PEM and MRI showed higher sensitivity (92.8%) for PEM for index lesions and slightly lower (85%) for additional lesions. MRI showed no significant difference in sensitivity for index lesions (95%) and higher (98%) for additional lesions. Specificity was higher for PEM (74% vs. 48%). Whole-body PET had the worst sensitivity (67.9%). [50]. Other studies continue to show high PEM sensitivity 90–96% and specificity 84–91%, still higher than MRI [51,52,53]. When tumor size comes ino play, evaluation between PEM and PET/TC found no difference in assessing lesions greater than 2 cm in diameter, while PEM had better resolution for <2 cm lesions, [54]. Although, to the best of our knowledge, no study has compared BSGI directly to PEM, the overall sensitivity and specificity of the two methods are comparable, although confirmation is needed.

## 5. Conclusions

BSGI is a useful imaging modality that, as a functional imaging modality, reaches where anatomical imaging does not. With sensitivity comparable to MRI but higher than MMG and US, it can aid physicians and surgeons in obtaining the best therapeutic approach for the patient as it may distinguish benign and malignant lesions with higher specificity. Although it involves a higher radiation burden for patients, many studies have been conducted with BSGI approaches in which the injected dose was lowered. Incorporating BSGI for breast lesion diagnosis increases overall sensitivity in the detection of cancer lesions.

## Figures and Tables

**Figure 1 cancers-14-04619-f001:**
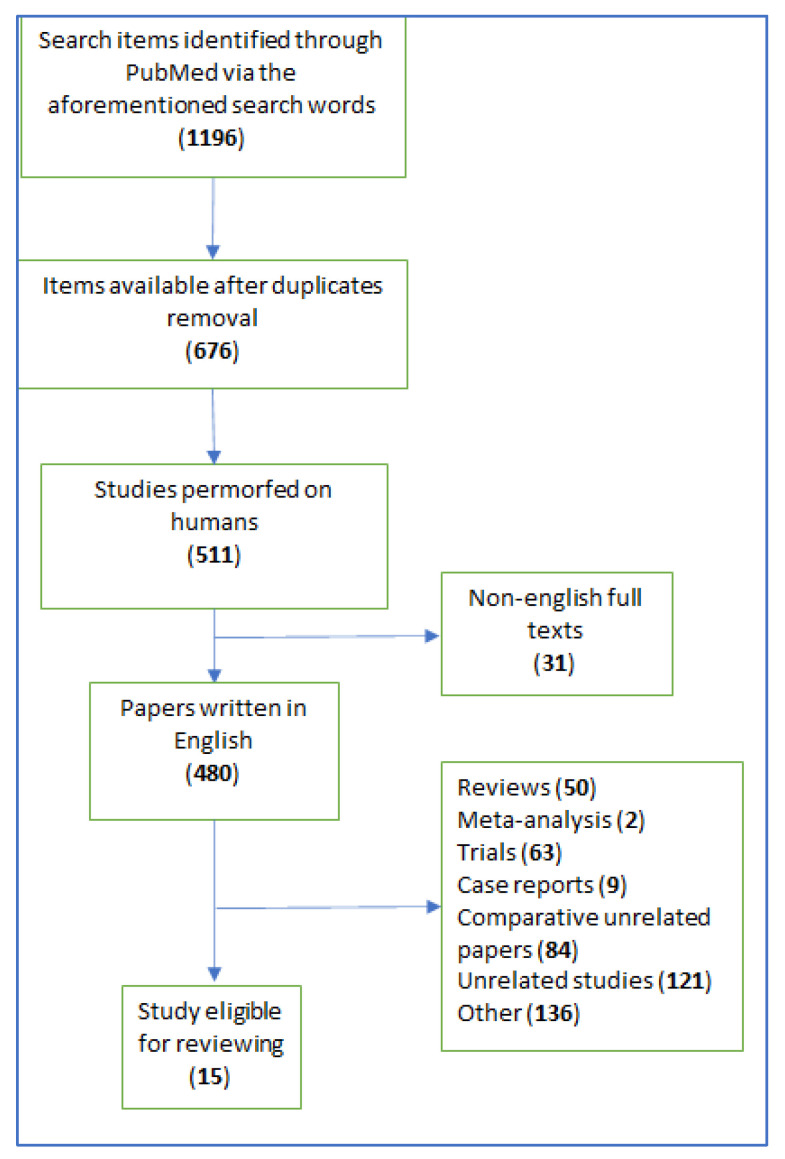
PRISMA flow-chart.

**Table 1 cancers-14-04619-t001:** A summary of studies that compared BSGI and MRI.

BSGI vs. MRI	Sensitivity	Specificity	PPV	NPV
Liu et al., 2020 [1]	91.7 vs. 92.5	80.7 vs. 69.7	87.1 vs. 86.5	87.2 vs. 81.5
Liu et al., 2021 [2]	76.9 vs. 83.9	70.6 vs. 58.8	Not esplicited	Not esplicited
Keto et al., 2011 [3]	89 vs. 94	Not esplicited	Not esplicited	Not esplicited
Brem et al., 2007 [4]	91 vs. 88	Not esplicited	Not esplicited	Not esplicited
Meissnitzer et al., 2015 [5]	90 vs. 88	56 vs. 40	85 vs. 80	67 vs. 56
Yu et al., 2016 [6]	80.35 vs. 94.06	83.19 vs. 67.69	87.1 vs. 81.9	75 vs. 88
Kim et al., 2014 [7]	68.6 vs. 91.4	Not esplicited	Not esplicited	Not esplicited
Kim et al., 2012 [8]	88.8 vs. 90.1	90.1 vs. 39.4	76.6 vs. 35.8	95.5 vs. 93.3
Kim et al., 2019 [9]	70.2 vs. 83.3	90 vs. 60	94.6 vs. 84.2	51.9 vs. 56.3
Brem et al., 2008 [10]	89 vs. 100	71 vs. 25	53 vs. 33	94 vs. 100

**Table 2 cancers-14-04619-t002:** A collection of papers in which BSGI was compared to mammography.

BSGI vs. Mammography	Sensitivity	Specificity	PPV	NPV
Liu et al., 2020 [1]	91.7 vs. 77.3	80.7 vs. 74.5	87.1 vs. 81.2	87.2 vs. 69.8
Lee et al., 2012 [11]	95.45 vs. 93.64	90.93 vs. 90.66	76.09 vs. 75.18	98.51 vs. 97.92
Liu et al., 2020 [12]	94.9 vs. 91.5	78.3 vs. 48.3	89.5 vs. 77.5	88.7 vs. 74.4
Cho et al., 2016 [13]	90.9 vs. 74.2	78.1 vs. 56.3	74.1 vs. 53.9	92.6 vs. 76.1
Brem et al., 2007 [4]	91 vs. 82	Not esplicited	Not esplicited	Not esplicited
Meissnitzer 2015 [5]	90 vs. 85	56 vs. 28	85 vs. 76	67 vs. 41
Yu et al., 2016 [6]	80.35 vs. 75.6	83.19 vs. 66.39	87.10 vs. 76.05	75 vs. 65.83
Kim et al., 2012 [8]	92.2 vs. 53.6	89.3 vs. 94.7	94.6 vs. 95.3	84.8 vs. 50
Tan et al., 2016 [14]	94.1 vs. 84.5	Not esplicited	Not esplicited	Not esplicited

**Table 3 cancers-14-04619-t003:** Sensitivity, specificity, PVV and NPV for BSGI and ultrasound in the included studies.

BSGI vs. Ultrasound	Sensitivity	Specificity	PPV	NPV
Liu et al., 2020 [1]	91.7 vs. 82.1	80.7 vs. 70.8	87.1 vs. 80	87.2 vs. 73.5
Lee et al., 2012 [11]	95.45 vs. 98.18	90.93 vs. 87.09	76.09 vs. 69.68	98.51 vs. 99.37
Liu et al., 2020 [12]	94.9 vs. 93.2	78.3 vs. 53.3	89.5 vs. 79.6	88.7 vs. 80
Cho et al., 2016 [13]	90.9 vs. 87.9	78.1 vs. 19.8	74.1 vs. 43	92.6 vs. 70.4
Meissnitzer 2015 [5]	90 vs. 99	56 vs. 20	85 vs. 77	67 vs. 83
Yu et al., 2016 [6]	80.35 vs. 82.14	83.19 vs. 77.31	87.10 vs. 83.64	75 vs. 75.41
Kim et al., 2012 [8]	92.2 vs. 91.5	89.3 vs. 53.3	94.6 vs. 80	84.8 vs. 75.5
Tan et al., 2016 [14]	94.1 vs. 84.5	Not esplicited	Not esplicited	Not esplicited

**Table 4 cancers-14-04619-t004:** QUADAS 2 score of all included studies.

	Risk of Bias Assessment				Applicability Concerns Assessment		
	Patient Selection	Index Test	Reference Standard	Flow and Timing	Patient Selection	Index Test	Reference Standard
Liu et al., 2020 [12]	Low	Low	Low	Low	Low	Low	Low
Liu et al., 2021 [24]	Low	Low	Low	Low	Low	Low	Low
Keto el al. 2011 [25]	Low	Low	Low	Low	Low	Low	Low
Brem et al., 2007 [26]	Low	Unclear	Low	Low	Low	Low	Low
Meissnitzer et al., 2015 [27]	Low	Low	Low	Low	Low	Low	Low
Yu et al., 2016 [28]	Low	Low	Low	Low	Low	Low	Low
Kim et al., 2014 [29]	Low	Low	Low	Low	Low	Low	Low
Kim et al., 2012 [30]	Low	Low	Low	Low	Low	Low	Low
Kim et al., 2019 [31]	Low	Low	Low	Low	Low	Low	Low
Brem et al., 2008 [32]	Low	Low	Low	Low	Low	Low	Low
Lee et al., 2012 [33]	Low	Unclear	Low	Low	Low	Low	Low
Lee et al., 2020 [34]	Low	Low	Low	Low	Low	Low	Low
Cho et al., 2016 [35]	Low	Low	Low	Low	Low	Low	Low
Tan et al., 2016 [36]	Low	Low	Low	Low	Low	Low	Low
Brem et al., 2016 [37]	High	High	Low	Low	High	High	Low

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
