# Peer review of "Breast-Specific Gamma Imaging: An Added Value in the Diagnosis of Breast Cancer, a Systematic Review"

_cancers, 2022, doi:10.3390/cancers14194619_

Round 1

Reviewer 1 Report

General Comments:

It has been well known that BSGI may have similar sensitivity of MRI but better specificity than MRI with controversial arguments of different interpretation, patient selection and relatively small case numbers. Since contrast enhancement as well as nonspecific radiopharmaceuticlas all imagings can not be tumor-specific. More tumor specific agents like estradiol, tamoxifen or herceptin may provide higher specificity. Your paper is fairly well written, but needs an English editorial assistance.

Specific Comments:

1. line 153-4: ....superior to...------>...greater than

2. line 154:..luminal,------->..luminal lesions,

3. lines 155, 170, 175, 180, 185, 187: need space between last sentences and references

4. lines 166, 197:...over------>greater than

5. line 177: Mean value-----> What kind of value? SUV?

6. line 183: Also Kim et al.------>Kim et al. also

7. line 212: Therefore, not only young women....------>Only young women..

8. line 233: Estimates------>Estimated

Author Response

All suggestions were accepted, and changes were made in the text

Reviewer 2 Report

Introduction:

Please use Bq, Sv, and Gy throughout as these are SI derived unites for activity/radiation dose

Page 2 line 75 - Is the dose to the breast correct? 2 Gy is a very high dose.

Methods:

Searching only one database would not secure a total overview of the field.

The methods description is not detailed enough, and a lot of essential methodological information is missing. Please use the Preferred reporting items for Systematic reviews and Meta-Analyses (PRISMA) checklist and flow diagram to secure all elements are reported for a systematic review/meta-analysis as this.

Author Response

All suggestions were accepted, and changes were made in the text

Figure 1. PRISMA flow-chart and  Table 4. QUADAS 2 score of all included studies were added.

Reviewer 3 Report

This is a comprehensive review of the literature comparing breast specific gamma imaging to standard breast imaging techniques. The authors clearly describe the advantages and limitations of the techniques compared. We can see that the main limit to the development of BSGI is the higher irradiation, especially compared to MRI.

A medico-economic study related to the extra cost of unnecessary biopsies generated by the lack of specificity of MRI should be interesting.

Do the authors have data to do this analysis ?

Author Response

As requested, a medical-economic study related to the additional cost of diagnostic tests was analyzed and reported under discussion

Round 2

Reviewer 1 Report

Your paper has been improved along the lines our reviewers commented.

Author Response

Thank you

Reviewer 2 Report

Good work on the changes

Author Response

thank you

Reviewer 3 Report

Authors added an interesting medical-economic analysis in the discussion. A comparison with PET/CT imaging (whole-body or mammi-PET) could be also interesting.

Author Response

A comparison with PET/CT imaging (whole-body or mammi-PET) was added